# Response to the Regulation of Video Games under the Youth Media Protection Act: A Public Health Perspective

**DOI:** 10.3390/ijerph19159320

**Published:** 2022-07-29

**Authors:** Suzanne Lischer, Emilien Jeannot, Lukas Brülisauer, Niels Weber, Yasser Khazaal, Samuel Bendahan, Olivier Simon

**Affiliations:** 1School of Social Work, Lucerne University of Applied Sciences and Arts, 6002 Lucerne, Switzerland; 2Addiction Medicine, Lausanne University Hospital (CHUV), 1004 Lausanne, Switzerland; emilien.jeannot@chuv.ch (E.J.); lukas.brulisauer@chuv.ch (L.B.); yasser.khazaal@chuv.ch (Y.K.); olivier.simon@chuv.ch (O.S.); 3Faculty of Medicine, Institute of Global Health, Chemin de Mines 9, 1202 Geneva, Switzerland; 4Consultation Psychothérapeutique de Montriond, 1006 Lausanne, Switzerland; contact@nielsweber.ch; 5Département de Comportement Organisationnel, Faculty of Business and Economics (HEC Lausanne), University of Lausanne (UNIL), 1015 Lausanne, Switzerland; samuel.bendahan@unil.ch

**Keywords:** policy, gaming, video games, gaming disorder, microtransactions, adolescents

## Abstract

The Swiss Youth and Media Act, which is about to enter into force, is an attempt to provide a legislative framework for video game use. Among other inclusions, the law intends to make providers more accountable by taking measures to protect minors from harm that can be caused by improper use of video games. However, it is a challenge to create a legal framework that can adequately regulate the evolving features of video games. Legislators must find a suitable regulatory approach which takes into account the fact that there is an increasing convergence between video games and gambling, particularly with the introduction of loot boxes. Moreover, there is a need for regulation, including the prohibition of misleading designs, the introduction of additional protection for minors, and the assurance of transparency of transactions. Appropriate policy legislation and consumer-protection measures are needed to protect people using these types of products, particularly children and adolescents. Further work should focus on assessing game characteristics to refine regulatory models to promote safe gaming. Based on experiences from the field of psychoactive substances as well as that of gambling, it is now a matter of developing a matrix of harm with elaborated categories: a tool that makes it possible to evaluate the potential harms of certain game design in an evidence-based manner.

## 1. Introduction

Today, 64.5% of the Swiss population plays video games at least several times per year [1]. This strong and very diversified practice raises the question of the need for regulation, especially in consideration of risky gaming behaviors and concerns related to gaming disorder [2,3]. The regulation response remains rather weak in Switzerland and elsewhere. The Swiss Youth and Media Act, which is about to enter into force, is an attempt to provide a legislative framework for video game use. Within this framework, the aim of the act is to better protect minors from inappropriate media content. Age labels and age controls for films and video games should be regulated uniformly throughout Switzerland [4]. Among other inclusions, the law intends to make providers more accountable by taking measures to protect minors from harm that can be caused by improper use of video games. This includes age-category systems and rules for age indications and controls. The integration of industry self-regulatory measures, such as the Entertainment Software Rating Board (ESRB) and Pan-European Game Information (PEGI), into the legal framework is also envisaged. The Swiss Youth and Media Act is based on the European Union’s Audiovisual Media Services Directive, which underwent a revision in late 2018. The debates regarding the law are an indication of the increasing complexity associated with media and, in particular, with video games. From the perspective of protecting minors, for example, there is the question of regulating microtransactions. Moreover, with the introduction of loot boxes (which are random rewards of uncertain value that can be purchased with real-world money) [5], legislators need to create appropriate regulation which takes into account that the lines between video games and gambling are blurring [6]. This new game feature has raised concerns about the high addictive potential of certain games [7]. However, the video game market is highly volatile and thus in a constant state of change. It is a challenge to create a legal framework that can adequately regulate the evolving features of video games. It is also likely that policymakers will not be able to follow such developments in detail. Thus, they may lack the knowledge base needed to determine measures that can adequately address the potential harms. This is mainly due to the fact that there is currently no assessment tool to evaluate the potential harm of video games based on their structural characteristics, as is the case for substances or gambling [3,8,9]. While video game content rating systems such as those of PEGI or the ESRB may help consumers to make informed decisions regarding their choice of video games, they only offer limited protection. 

Based on the current state of knowledge, this viewpoint proposes to identify and summarize games (as a product) and the internal mechanisms that may pose a risk for problem gambling behavior. By describing different forms of video games and their inherent mechanics, the present viewpoint aims to ensure that appropriate legislation can be enacted through this means in Switzerland, as well as in other jurisdictions worldwide. By highlighting the possible harms that can result from the features of video games, a recommendation will be made as to how preventive aspects can be adequately considered in regulation. Moreover, by presenting what implications may result from these internal characteristics, it emphasizes the need for future research concerning the development of a valid assessment grid that would assist policymakers in refining government regulation of video game products. 

Notwithstanding the endeavors described, there is little epidemiological data available on gaming in the general population in Switzerland or elsewhere. A Swiss study from 2015 on “problematic Internet use” (i.e., many different online activities, including gaming) reports that the prevalence in the general population over the age of 15 is 3.8%, with the most affected age group (15–24 years) having a prevalence rate of 11.2% [10] (Table 1). However, there are no national or international data on help-seeking for gambling disorders [11]. Nonetheless, descriptions of the activities of some of the dedicated support facilities have been published, showing a specific increase in help-seeking in relation to gaming disorders [11,12,13,14]. 

## 2. Problematic Gaming Behavior 

While there can be many benefits associated with gaming [18], there are concerns that, in addition to game content (e.g., sex, violence, substance use, offensive language, etc.), overuse can be a real issue [19]. Although video games are an enjoyable activity for most people, consumers with various intrapersonal and interpersonal risk factors may be tempted to use computer games as a strategy for handling personal problems [20]. For instance, some people may engage in gaming to escape from real-life problems [21] and may neglect relationships, school or work-related duties, and even basic physical needs [19,20]. Video gaming can thus be conceptualized as belonging to a continuum, which ranges from an enjoyable activity to a pathology where it is used to express psychological malaise [22]. Therefore, video gaming problems have attracted considerable attention in recent years and are now recognized as behaviors that may be subject to addiction. Gaming disorder (GD) was officially adopted at the World Health Assembly in May 2019 as a diagnosis in the 11th edition of the International Classification of Diseases [23]. The ICD-11 diagnostic criteria for gaming disorder include impaired control over gaming, increasing priority of (and preoccupation with) gaming, and continuation or escalation of gaming despite experiencing negative consequences [23]. The behavioral pattern of gaming must be related to significant impairment in personal, family, social, educational, occupational, and/or other important life domains for a duration of 12 months. Functional impairment is a central criterion in many mental health disorders, such as gambling and gaming disorders [24]. 

Problematic video game playing is multifaceted rather than a unitary phenomenon. As a result, multiple factors emerge in different ways and at different levels of analysis (e.g., psychological, biological, social, situational, or structural). The essence of this view is that no single level of analysis is sufficient to explain the emergence or maintenance of video-game-playing behavior [25]. However, in order to understand what motivates player behavior, it is necessary to study games’ (as a product) internal mechanisms [3]. In fact, microtransactions (of which loot boxes are a part), are not game mechanics per se, but are mainly economic mechanics thought to make players spend more money [26]. We must therefore take a closer look at the way the industry integrates them into video games.

## 3. Structural Characteristics of Video Games

Video games are complex art forms that combine elements of storytelling, graphics, sounds, game mechanics, and sometimes ubiquitous elements into digital artifacts [27]. The precise definition of “video game” can be debated within the scientific community. The Swiss legislators did not consider it necessary to propose a definition in the text of the law. However, it is generally agreed that this is an activity involving play, which is practiced on a screen, alone or with others, making use of skills and sometimes of chance. Video games can be classified according to genres, themes and types [28]. *Genres* are defined by the gameplay mechanics, for example battle royale (BR) games, first-person shooter (FPS) games, multiplayer online battle arena (MOBA) games, and real-time strategy (RTS) games. Video game *themes* are categorized based on how in-game visual contents look (are presented) [28]. Video games can be played across a wide variety of platforms, including personal computers; handheld devices such as mobile phones, tablets, or the Nintendo Switch; as well as game consoles, such as the PlayStation 5, which are used with a television [18]. Video game *types* are therefore used to classify video games based on the specific combination of electronic components or computer hardware (for example arcade, computer, console, handheld, and mobile) [28]. Each of these genres or types is likely to host microtransactions, depending on the business model chosen. Furthermore, in spite of a lack of adequate assessment of in-game related processes, it seems that punishment and reward allocation models, social characteristics, and monetization patterns have been associated with higher scores on gaming disorder measures [3]. 

## 4. Business Models of Video Games 

The structure of video games depends also on the business model. Therefore, it is worth taking a closer look at the different business models, especially because the introduction of monetization strategies has changed traditional game design [29]. On the one hand, a distinction can be made between the different models of accessibility to customers (pay-to-play versus free-to-play). On the other hand, there are several revenue-model options for video games, and the game provider can select their preferred model depending on the channel through which the product or service provides value to the company. The potential schemes can be classified as: coin-op, retail, digital distribution, advertising, subscription, microtransactions, and player-to-player commerce [28]. At the beginning of video gaming history, the user had to pay the supplier (publisher) to buy the game and use it after installing it on their device [30]. The pay-to-play (P2P) business model is a traditional model in the video game industry that requires customers to make a payment before getting full access to the functional content of a product or service [28]. However, over time, the game market became more and more saturated, and it also became easier to download an illegal copy of a game shortly after its release. This prompted publishers to modify the model and share the games in order to make a continuous profit. Therefore, rather than selling the full version of the game, the basic version was made available at a low price or even for free, while all add-ons and premium features became available to buy as additional game features. Such purchase options create a lack of transparency regarding the average costs of the game in relation to future purchases. In gaming terminology, games available free of charge are generally referred to as free-to-play (F2P), or freemium, whereby income is generated by selling small “pieces” of the game [30] (the so-called microtransactions). The F2P business model, which thus seems to be an attractive means of monetizing games, has been adopted by new developers in the video game industry. Some of these providers are generating a few million U.S. dollars (or the equivalent) in revenue [31,32]. Nowadays, online multiplayer games are increasingly based on the F2P model, and even retail (P2P) games feature in-app purchases to gather extra revenue on top of the fixed price [33]. This is the emergence of the concept of the game as a service. The product continues to be fed with content, sometimes for years after its release [31].

## 5. Microtransactions

Games with a payment feature have increasingly raised concerns due to the psychological mechanisms that induce payment in such games. Attention is also drawn to the increasing problems of payments made by children, which, according to some authors, are a risk factor for the future onset of gambling addiction [30]. Therefore, a closer look at the providers’ monetization strategies is indispensable. Microtransactions are defined as in-game payments for items or unlockable content made directly with real-world money or indirectly through the buying of virtual currency [34]. Many microtransactions allow players to buy decorations and alternative costumes or items that offer no advantage in the game and have a purely aesthetic nature. For example, in the multiplayer battle royale game *Fortnite* (Epic Games, 2017)*,* players can spend real-world money to buy in-game “emotes” that allow them to express ideas and feelings via the movements of their in-game avatar [35]. 

However, not all microtransactions in games are purely cosmetic. Players now also have the option of buying virtual items and bonuses that increase their chances of success in the game [35]. These payments do not serve as a prerequisite for participation in the game. Instead, they enable the paying customer to have virtual benefits within the game. The benefits can include aesthetic advantages, stronger avatars (user profiles), shortened waiting times, or advancement to a new level. When these elements dominate over the skill components of the game, the games are called pay-to-win (P2W) games [36]. 

P2W games are thus defined as those video games that offer the possibility of spending money on content, items, or events that help the player to progress in the game. While the optional purchase of in-game items and events is a common feature of online games, P2W products are functional in that they give players a distinct advantage in the game and increase the likelihood of winning. More recently, however, P2W games have been criticized by both developers and players for both causing frustration and being perceived as unfair [37]. A new phenomenon in digital games further introduces randomness into the purchase of in-game items: so-called loot boxes. Loot boxes are items in video games that can be purchased for real money but offer players a random reward of uncertain in-game value. Thus, when players pay their money, they cannot know exactly what return they will get for their investment [38]. A loot box can be a P2W product if its content has functional properties that enhance players’ probability of winning [37]. The game *Diablo Immortal*, released at the beginning of June 2022 on iOS and Android, illustrates the problem. This free-to-play game (that is to say free at the beginning) puts loot boxes at the heart of its gameplay: it is indeed not possible to go beyond a certain stage in the game without buying loot boxes. The YouTube channel Bellular News [39] estimated that to progress your character to the maximum level, you would have to spend about USD 110,000. In accordance with existing gambling legislation, this game has been banned in Belgium and the Netherlands [40]. 

## 6. What Structural Characteristics Cause Harm?

As participants in a for-profit industry, the objective of game developers and publishers is to create attractive and entertaining games. This implies that some vendors strive to produce games that will be played by as many players as possible for as long as possible, as this increases the game industry’s revenue streams. To achieve repeated play, developers use elements of game design based on psychological mechanisms such as variable reinforcement (e.g., receiving valuable in-game rewards at irregular and unpredictable intervals) that encourage long-term player investment [19]. The literature from the last decade on problematic gaming has largely focused on massively multiplayer online role-playing games (MMORPG) [41]. Most of the studies available considered overuse of MMORPGs within the framework of the concept of “online games addiction” (generally defined as the tendency to spend an excessive amount of time on online games while displaying several symptoms of pathological behaviors, such as loss of control, cravings, and intrapersonal/extrapersonal problems) [42]. Almost all the literature includes a methodological bias, as video game addiction had not yet been qualified by a reference diagnosis. However, supposing that monetarization strategies are creating a consumer commitment towards the game, it is reasonable to assume that free-to-play games might also bear an addictive potential [29]. Particularly, they might encourage favouring play over personal obligations, or lead to financial problems. 

The paradox is that microtransactions are not game mechanics in themselves but monetization tools (buying is not playing). Thus, the expansion of microtransactions in games raises some concerns that certain players (e.g., younger users) may be vulnerable to excessive spending or impulsive purchases. There is also a suggestion that some activities may provide a path to gambling and/or that some games are designed to resemble slot machines and have the same basic element of chance governing the distribution of in-game rewards. At the same time, there are concerns that some in-game purchase systems that use randomness, such as loot boxes, may contribute to excessive gaming behavior and psychological overinvestment in video games in general [7]. Accordingly, loot boxes are taken to be a feature of modern video games which are claimed to represent “predatory monetization” and are an example of the so-called “gamblification” of games [43]. However, the question of whether loot boxes are gambling is controversial [44]. Though legal definitions vary, three key components are commonly necessary for bona-fide gambling: “Consideration” refers to participants’ risking something of value on the activity; “chance” describes the element of uncertainty in the activity; and the “prize” is the potential reward or outcome [44,45]. In the context of the discussion about whether loot boxes are gambling, the crucial issue is the price element because the items in the microtransactions are virtual items, and the question is unresolved as to whether virtual goods have a “real” price. Drummond and colleagues, for example, argue that these items do have value for players, even if they are virtual goods [45]. Given the apparent overlap between loot boxes and gambling, this raises questions about the relationship between gambling-related addiction and loot box use. With regard to this issue, Zendle and Cairns examined whether specific characteristics of different loot boxes strengthened the association between loot box spending and problem gambling among adult gamblers. The results provide evidence that this association is strengthened when the contents of the loot boxes can be exchanged for real money. In addition, it is strengthened when loot boxes show “near-losses” of things players could have won and when the amount players spend on loot boxes is hidden behind in-game script purchases [46]. Studies have shown that the more money gamers spend on loot boxes, the more severe their problem gambling is. This association appears to be stronger in adolescents than in adults, which is concerning [47]. Thus, a study which considered adolescents and young adults aged 16–24 years indicated that the purchase of loot boxes was highly associated with problem gambling, the strength of this association being of similar magnitude to gambling online on casino games or slots. The authors concluded that young adults purchasing loot boxes within video games should be considered a high-risk group for the experience of gambling problems [48]. 

## 7. Industry Self-Regulation

In April 2020, the Entertainment Software Rating Board (ESRB) and Pan-European Game Information (PEGI) introduced the Interactive Element of “In-Game Purchases (Includes Random Items)” and the Content Descriptor of “Includes Paid Random Items”, respectively. Both argued that these measures are intended to better inform consumers by flagging the random nature of certain in-game purchases Accordingly, these labels would ensure consumer protection. [49,50]. Whether this self-declaration is sufficient to prevent potential harm to minors or vulnerable groups is debateable. For example, is the term “random articles” sufficient to provide consumer protection? In particular, it does not convey sufficient concrete information about the mechanisms involved to allow consumers to effectively identify them and make an informed choice. This is in stark contrast to established descriptions such as “violence” and “gambling”, which have inherent meanings. The meaning of “Random Items” is not explained alongside the labels. This forces consumers to consult the organizations’ websites for explanations. Families with vulnerability factors pay little attention to these pictograms, in any case. Moreover, as game boxes disappear in favor of digital versions, the symbols become even less accessible. Critics also caution that these measures do not inform consumers when random items are available for purchase, how much they cost, the probability of winning, whether they affect gameplay or are merely cosmetic, or whether they can be redeemed (i.e., converted from in-game virtual items to real currency) [51]. 

## 8. Discussion 

From a public health perspective, the regulation of loot boxes and other microtransactions is highly pertinent, though it must of course acknowledge the commercial interests of game developers and the gaming industry. In addition, it will be important to take into account the tension between the conflicting interests of all involved parties [52]. However, microtransactions have raised concerns because of the financial costs incurred by children and adolescents, while loot boxes represent the peak of the iceberg, because these features evade regulations, therefore permitting underage gambling and its promotion [53]. The relationship between problem gambling and spending on loot boxes is of particular concern because loot boxes are often found in games played by children and young people: More than half of the top-selling mobile games now contain loot boxes. No less than 94% of these games are considered suitable for children aged 12 and older. Given that young people are more prone to impulsive behavior and find risk-taking more appealing than other age groups do, the prevalence of loot boxes in games that are popular with teenagers is particularly worrying [54]. 

To effectively address the above challenges, researchers need to understand and consider the characteristics and functioning of in-app purchases and gambling features, including how children are drawn in and which subgroups of players are most vulnerable. Thus, video games should be regulated according to classification systems that purport to relate to the harms and risks of each game. However, the methodology and processes underlying classification systems are generally not specified or made transparent, which reduces confidence in their accuracy and undermines health education messages [8]. Therefore, based on experiences from the field of psychoactive substances as well as that of gambling, it is now a matter of developing a matrix of harm with elaborated categories [8,9]: a tool that makes it possible to evaluate the potential harms of certain game design in an evidence-based manner. This matrix of harm could also consider other sources of harm, such as inappropriate content or whether structural conditions are in place that protect against harassment and bullying.

Another key area of investigation is parental knowledge, as it enables effective prevention methods [53]. However, we cannot base a solid strategy on simply empowering parents. They need much more help, and this requires much more coherent, solid, and uniform information. Therefore, advocates and policymakers must attempt to hold the gaming industry accountable and act in a socially responsible manner to advocate for the welfare of children and families, and the sustainability of responsible gaming [53]. It is up to politicians to adopt appropriate measures to put them into practice, based on sound science and practical experience from the field. As the following example illustrates, there are ways in which consumers and public health experts can get involved in shaping the regulatory framework: In June 2022, European consumer protection organizations called for stricter regulation, including the prohibition of misleading designs, the introduction of additional protection for minors, and the assurance of transparency of transactions. Furthermore, the publisher EA Games was asked to indicate the prices of the content it sells in its video games in euros and not only in the form of game currency [55]. There are also voices in the Swiss parliament that want to force publishers to display the actual prices of microtransactions [56]. Appropriate policies for consumer protection, psychologically informed interventions, and ethical guidelines for the design of games are needed to protect the well-being of consumers. This is especially true for young people, who are particularly vulnerable, being the most avid players and yet the most poorly informed subgroup of consumers [57]. 

Reviews [52,58] indicate that regulatory models around the world differ in their approaches to classifying loot boxes as gambling and regulating microtransactions. The most frequently discussed regulatory frameworks concern Belgium and the Netherlands, which, given the specifics of their legal frameworks for online gambling, have banned the use of certain loot boxes in their territories. China, for its part, has required publishers to better inform consumers by indicating the likelihood of obtaining the desired virtual goods via the relevant loot boxes [59]. However, the body of research on a feasible regulatory framework for video games from a public health perspective is still sparse. Further research into this matter would be useful.

In Switzerland, the legal draft of the Federal Act on the Protection of Minors in respect of Film and Video Games is still pending. The responsible commission decided in May 2022 to remove all regulation of microtransactions as well as a paragraph on prevention and media literacy promotion from the proposed draft. This does not seem consistent to us, given the situation as set out above and the previous debates. Admittedly, microtransactions are not game mechanics. However, the video game, as a consumer product, gives access to these microtransactions. If it is coherent to separate them from a semantic point of view, it would be unwise not to legislate for this facility. However, the consultation process has not been concluded yet.

## 9. Conclusions

This viewpoint intends to summarize the evidence on the potential harm of certain video games (as a product) and internal mechanisms as well as inherent commercial technologies that may pose a risk for problem gaming behavior. It thus aims to inform policymakers who can use this contribution to develop adequate regulation. Given the prevalence of video games, it is surprising that they are barely regulated in Switzerland. The fact that the Swiss legislature now intends to regulate video games within the Swiss Youth and Media Act is therefore to be seen as positive. 

The Swiss Youth and Media Act should create a legal framework that protects children and adolescents from aggressive commercial techniques used by the video game industry. They must be protected from game design elements based on psychological mechanisms designed to encourage players to make long-term investments. The above explanations indicate that some video games can provide a pathway to gambling, as some games are designed to resemble slot machines and incorporate the same basic random choices for in-game rewards. What must therefore be emphasized at this point is that: children and adolescents should be restricted from accessing any mechanisms which fit a jurisdiction’s legal definition of gambling or which fulfil all the criteria of the psychological definition of gambling [52]. Furthermore, loot boxes are purchased with a virtual currency, which in turn is purchased with Swiss francs. The price of a loot box in Swiss francs is therefore not known to a player or is only learned with great effort. In order to prevent children and adolescents from excessive spending or impulsive purchasing, the price must be displayed in Swiss francs.

Even if this point of view is obvious to everyone (or at least it should be), the regulation of the Youth Media Protection Act will remain a challenge. It appears that the model proposed in the draft law is based on the regulation of video content, which is not sufficiently adapted to the development and interactivity of the mechanisms of video games. However, the problem of loot boxes would be a clear risk factor amenable to regulation. 

The fact that Swiss legislators take rating boards such the Entertainment Software Rating Board (ESRB) and Pan-European Game Information (PEGI) into account is reasonable and important. From a prevention perspective, it would make sense if the expertise of youth media protection associations were also considered for the classification of the respective video games. 

In addition, to ensure compliance with youth protection regulations, test purchases are required. This obligation should be detailed at the legislative level. However, due to the difficulties in regulating purchases made online (and therefore at home), other strategies must also be explored, and in particular measures related to providing consistent and precise information. It is not possible to have an imprecise professional or political discourse on these issues.

As a basis for regulation, large-scale, international, representative studies are needed to provide evidence on the prevalence and adverse health consequences of video gaming [53]. Further work should focus on assessing game characteristics to refine regulatory models to promote safe gaming. The funding of research is a governmental task and should be supported by expert scientific funds at the best level of practices guaranteeing independence and transparency. In terms of future prospects, it seems increasingly important that the health and social fields, including research, investigate the subject of social engineering more closely. To better detect what may be problematic in such games, we also need to listen to and collaborate with professionals who are familiar with algorithms, game design, and consumer incentives. This will allow us to better target what needs to be legislated. 

## 10. Terminology

**Battle Royale (BR) game:** A particular genre of video game that mixes survival and shooting games, based on the so-called “last man standing” mechanic.

**First-person shooter (FPS) game:** Shooter video game centred on weapon-based combat in which the player experiences the action through the eyes of the character, from a first-person perspective. 

**Free-to-play (F2P) game:** Video game which can be obtained free of charge.

**Gaming disorder (ICD11):** Characterized by a pattern of persistent or recurrent gaming behavior (“digital gaming” or “video-gaming”), which may be online (i.e., over the internet) or offline, manifested by: 1. impaired control over gaming (e.g., onset, frequency, intensity, duration, termination, or context); 2. increasing priority given to gaming to the extent that gaming takes precedence over other life interests and daily activities; and 3. continuation or escalation of gaming despite the occurrence of negative consequences. The gaming behavior and other features are normally evident over a period of at least 12 months in order for a diagnosis to be assigned. 

**Loot box:** In-game item or reward paid for with real-world money that contains randomized contents whose value is unclear at the time of purchase.

**Massively multiplayer online role-playing game (MMORPG):** Role-playing game in which the player assumes the role of an avatar and takes control over many of its actions. This video game contains a social aspect, with players being able to interact with one another, and is characterized by a persistent world which keeps existing and evolving while the player is offline.

**Microtransactions:** In-game payments for items or unlockable content made directly from real-world money or indirectly through the buying of virtual currency.

**Multiplayer online battle arena (MOBA) game:** Strategy video game distinguished by two teams of players competing against each other on a battlefield with the objective of destroying each other’s main structure. Each player controls a single character with the aim of improving its abilities over the course of a game, thus contributing to the team.

**Pay-to-win (P2W) game:** Video game that offers the player the possibility to spend money on content, items, or events which serve to advance the game.

**Real-time strategy (RTS) game:** Strategy video game in which each player places structures and manipulates multiple units in order to secure areas of the map and/or attack their opponents’ assets. It usually features specific aspects such as base-building, resource-gathering, and indirect control of units.

## Figures and Tables

**Table 1 ijerph-19-09320-t001:** Epidemiological data.

	In Switzerland	Worldwide
**Number of gamers**	65.4% of the population played video games at least several times per year in 2021 [1].	In 2020, nearly 3.1 billion people worldwide played video games, representing about 40% of the world population [15].
**Revenues derived from games market**	Revenue is projected to reach USD 421.10 million in 2022 [14].	USD 175.8 billion generated in 2021 [16]
**Prevalence of gaming disorder**	3.8% [9]	Between 1.96% and 3.05%, depending on sampling and assessment criteria [17]

## Data Availability

Not applicable.

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
