# Peer review of "Response to the Regulation of Video Games under the Youth Media Protection Act: A Public Health Perspective"

_ijerph, 2022, doi:10.3390/ijerph19159320_

Round 1

Reviewer 1 Report

This is a well written paper describing the characteristics of gaming design that are relevant to regulations which aims to prevent harms from video games in youth. It is an important discussion to have as Legislation is being considered in Switzerland. The paper embeds comments about the harms as these structural elements are described. The paper may, as an alternative, have covered the topic with sections beginning with the harms and then described the mechanisms. However, the actual legislation needs to produce regulations that deal with the game design and sales so the way it is structured is appropriate. 

It is quite dense with information. In the reading, the detail about the technical aspects of gaming sometimes obscures the points about the harms this translates to. Although this is a paper for a peer reviewed journal, one hopes that it will be read by some of those who are influencing laws. Gaming is to some extent bringing about a paradigm shift in what we see as gambling. Legislators must struggle conceptually with the area so it is worthwhile ensuring it is easily understood. To this end, I think the paper could be strengthened with some concluding simple sentences capturing the game design elements and corresponding harms again. Please consider some simple statements to the effect - legislation can be applied to protect against the harms of: exposure to inappropriate content; financial exploitation (mainly through manipulations of micro-transactions etc); gaming addiction (through purposefully addictive design akin to the electronic gambling or ‘gaming’ machine approaches); intra-gaming gambling problems (predominantly via loot box design); gateway to gambling problems (this is not always just because of loot boxes) (Molde H et al 2018) etc. 

I also encourage the authors to include comment about what is known about legislation from other international jurisdictions (Western/WEIRD settings) and also a call for scoping research into this. Finally, I feel that it is relevant to consider how consumer (gaming disorder) and carer input could make its way into the legislation debate. 

Author Response

Comment 1: This is a well written paper describing the characteristics of gaming design that are relevant to regulations which aims to prevent harms from video games in youth. It is an important discussion to have as Legislation is being considered in Switzerland.

Answer 1: Thank you very much for your positive feedback and helpful suggestions. We have done our best to update the paper, in keeping with your comments.

Comment 2: … the paper could be strengthened with some concluding simple sentences capturing the game design elements and corresponding harms again.

Answer 2: Thank you for your suggestion. In the "Conclusion" section, we now make reference  to the game design elements and corresponding harms, and we make recommendations for the policymakers, which draw upon our previous explanations.

Comment 3:  … also encourage the authors to include comment about what is known about legislation from other international jurisdictions (Western/WEIRD settings) and also a call for scoping research into this.

Answer 3: We have addressed this important suggestion by referring to two reviews that have compiled the various regulatory models around the world. Specifically, we refer to the most frequently cited models from Belgium, the Netherlands and China. We also call for further research in this area.

Comment 4: I feel that it is relevant to consider how consumer (gaming disorder) and carer input could make its way into the legislation debate. 

Answer 4: The Discussion section now provides two specific examples of how civil society can get involved in the regulatory process to better protect consumers.

Reviewer 2 Report

The viewpoint describes an interesting and timely issue. The use of video games seems to be a debate in the literature. One should have entertainment and video games surely are good candidate. However, engaging in too much video games may cause problems in social functions and daily living. Therefore, it is essential to understand how we can best use of video games to improve people's mental health instead of causing problems in mental health. I appreciate that the authors provide their viewpoint to the regulation of video games among Swiss. I read this viewpoint with interest. However, some revisions could be made.

1. I think that at the beginning of t he Introduction, the authors should clarify different forms of video games and how the authors consider the relationships between online gaming and video games. This also indicates that whether the regulation has defined "video games". Also, what is the definition of video games in the present viewpoint?

2. Table 1. Some comparisons are hard to make. For example, the authors reported 65.4% for Switzerland and 3.24 billion worldwide; revenues for Switzerland in 2022 and those for worldwide in 2021. 

3. Under the section of "2. Problematic gambling behavior", the authors did not mention anything on gambling.

4. The authors described different forms of video games. However, I did not see how these associated with the regulation in Swiss. Also, it is unclear how these different forms of video games associated with mental health problems. I think that these two are the critical points in the present viewpoint. 

Author Response

Comment 1: I appreciate that the authors provide their viewpoint to the regulation of video games among Swiss. I read this viewpoint with interest.

Answer 1: Thank you very much for your interest in this article and for the opportunity to improve it by responding to your comments.

Comment 2; I think that at the beginning of the Introduction, the authors should clarify different forms of video games and how the authors consider the relationships between online gaming and video games. This also indicates that whether the regulation has defined "video games". Also, what is the definition of video games in the present viewpoint?

Answer 2: In the introduction, we now discuss what the legislators mean by video games. Furthermore, we present a definition of the term, upon which this viewpoint is based.

At the end of the text, we provide terminology related to  the different forms of video game. Following your suggestion, we also now refer to the issue of terminology in the introduction.

Comment 3: Table 1. Some comparisons are hard to make. For example, the authors reported 65.4% for Switzerland and 3.24 billion worldwide; revenues for Switzerland in 2022 and those for worldwide in 2021. 

Answer 3: We have added a source that gives a percentage value for game participation. This ensures comparability with Switzerland.

Comment 4: Under the section of "2. Problematic gambling behavior", the authors did not mention anything on gambling.

Answer 4: We have indeed made a mistake here. Instead of “gaming”, we used the term “gambling” in the title. We have corrected the error. The connection with “gambling” will be discussed below.

Comment 5: The authors described different forms of video games. However, I did not see how these associated with the regulation in Swiss. Also, it is unclear how these different forms of video games associated with mental health problems. I think that these two are the critical points in the present viewpoint. 

Answer 5: in the introduction section, we explain, why the knowledge of different form of video games are important with regard to the regulation in Switzerland. We also explain the association of video game features with mental health problems. We hope that the relevant associations have now been clarified

Round 2

Reviewer 2 Report

I am satisfied with the revised manuscript and have no more comments. I thank the reviewers for taking my comments to improve their work.